# Identification of Novel Candidate Genes and Variants for Hearing Loss and Temporal Bone Anomalies

**DOI:** 10.3390/genes12040566

**Published:** 2021-04-13

**Authors:** Regie Lyn P. Santos-Cortez, Talitha Karisse L. Yarza, Tori C. Bootpetch, Ma. Leah C. Tantoco, Karen L. Mohlke, Teresa Luisa G. Cruz, Mary Ellen Chiong Perez, Abner L. Chan, Nanette R. Lee, Celina Ann M. Tobias-Grasso, Maria Rina T. Reyes-Quintos, Eva Maria Cutiongco-de la Paz, Charlotte M. Chiong

**Affiliations:** 1Department of Otolaryngology—Head and Neck Surgery, School of Medicine, University of Colorado Anschutz Medical Campus, Aurora, CO 80045, USA; tori.bootpetch@cuanschutz.edu; 2Center for Children’s Surgery, Children’s Hospital Colorado, Aurora, CO 80045, USA; 3Philippine National Ear Institute, University of the Philippines (UP) Manila–National Institutes of Health (NIH), Manila 1000, Philippines; tlyarza@up.edu.ph (T.K.L.Y.); mlct19976@hotmail.com (M.L.C.T.); tgcruz1@up.edu.ph (T.L.G.C.); alchan@up.edu.ph (A.L.C.); mtreyesquintos@up.edu.ph (M.R.T.R.-Q.); 4Newborn Hearing Screening Reference Center, UP Manila—NIH, Manila 1000, Philippines; 5Department of Otorhinolaryngology, UP Manila College of Medicine—Philippine General Hospital (UP-PGH), Manila 1000, Philippines; 6Department of Genetics, University of North Carolina, Chapel Hill, NC 27599, USA; mohlke@med.unc.edu; 7Department of Anesthesiology, UP Manila College of Medicine, Manila 1000, Philippines; mcperez6@up.edu.ph; 8Office of Population Studies and Department of Anthropology, Sociology and History, University of San Carlos, Cebu City 6000, Philippines; nanette_rlee@yahoo.com; 9MED-EL, 6020 Innsbruck, Austria; celina.tobias-grasso@med-el.com; 10Institute of Human Genetics, UP Manila—NIH, Manila 1000, Philippines; eccutiongcodelapaz@up.edu.ph; 11Philippine Genome Center, UP Diliman Campus, Quezon City 1101, Philippines; 12UP Manila College of Medicine, Manila 1000, Philippines

**Keywords:** anomalies, *CBLN3*, cochlear implant, enlarged vestibular aqueduct, *GDPD5*, genetic testing, hearing loss, inner ear, *IST1*, malformations, temporal bone

## Abstract

*Background:* Hearing loss remains an important global health problem that is potentially addressed through early identification of a genetic etiology, which helps to predict outcomes of hearing rehabilitation such as cochlear implantation and also to mitigate the long-term effects of comorbidities. The identification of variants for hearing loss and detailed descriptions of clinical phenotypes in patients from various populations are needed to improve the utility of clinical genetic screening for hearing loss. *Methods:* Clinical and exome data from 15 children with hearing loss were reviewed. Standard tools for annotating variants were used and rare, putatively deleterious variants were selected from the exome data. *Results:* In 15 children, 21 rare damaging variants in 17 genes were identified, including: 14 known hearing loss or neurodevelopmental genes, 11 of which had novel variants; and three candidate genes *IST1*, *CBLN3* and *GDPD5*, two of which were identified in children with both hearing loss and enlarged vestibular aqueducts. Patients with variants within *IST1* and *MYO18B* had poorer outcomes after cochlear implantation. *Conclusion:* Our findings highlight the importance of identifying novel variants and genes in ethnic groups that are understudied for hearing loss.

## 1. Introduction

Hearing loss remains a public health burden worldwide, with global measures of the effects of hearing disability remaining steady over the past three decades [1]. With the use of sequencing technologies in the clinical setting, identification of genetic variants that predispose to congenital or early childhood hearing loss is becoming more accessible to a larger segment of the world population. When partnered with newborn hearing screening, massively parallel DNA sequencing holds the promise of identifying the genetic cause(s) of hearing loss at the earliest stage and can therefore guide the clinician in diagnosing and treating comorbidities, planning rehabilitative options such as hearing aids or cochlear implantation (CI), and when they become available, applying gene therapies [2,3]. Genetic hearing loss is a highly heterogeneous disease both in terms of clinical presentation and pathogenic DNA variants, which are usually rare and may lie within any one of hundreds of genes [2,4]. The identification of variants for hearing loss and their corresponding clinical profiles in patients from various populations will contribute to the large body of knowledge that is required to improve the utility of clinical genetic screening for hearing loss. A large community of clinicians and scientists continues to identify novel genes and variants for syndromic and nonsyndromic hearing loss. In the past two years alone, variants within novel hearing loss genes including *SLC9A3R1*, *ANLN*, *FOXF2*, *TOP2B*, *PLS1*, *PISD*, *CLRN2*, *AP1B1*, *SCD5*, *GGPS1*, *SLC12A2*, *THOC1* and *GREB1L* were identified in patients of various ethnicities [5,6,7,8,9,10,11,12,13,14,15,16,17,18,19,20,21,22]. To date, some of these genes remain candidates that require replication in additional hearing loss families and patients [6,9,14,19].

Compared to hundreds of known hearing loss genes, studies on the genetic causes of temporal bone malformations are limited, with only a few genes identified so far, to name a few: EVA and/or Mondini dysplasia and *SLC26A4*; superior semicircular canal dehiscence (SSCD) or posterior semicircular canal dehiscence (PSCD) and *CDH23*; and variable cochleovestibular anomalies in some patients with variants in *GJB2*, *POU3F4*, *SOX10*, *CHD7*, *SIX1* and *GREB1L* [20,21,22,23,24,25,26,27,28,29]. Prior knowledge of temporal bone malformations is important not only to prepare the surgeon for potential complications during CI but also to prognosticate outcomes after surgery [29,30,31]. Because CI is performed as early as three months old, occurrence of temporal bone anomalies might also be predicted earlier if genetic testing is performed at neonatal stage.

We previously studied a cohort of Filipino patients with hearing loss requiring CI [32,33,34]. In this cohort, we identified variants in known hearing loss genes in half of the patients, including a recurrent variant *SLC26A4* c.706C>G (p.Leu236Val) that was associated with bilaterally enlarged vestibular aqueducts (EVA) [32,33,34]. Of eleven patients with previously identified non-*SLC26A4* variants, only three had inner ear anomalies, including: EVA in a patient with an *EYA4* variant; and SSCD in one patient with a *KCNQ4* variant, and in another patient with CHARGE syndrome due to a *CHD7* variant [32,34]. On the other hand, of the genetically unsolved cases, 75% had temporal bone anomalies (Table 1) [33]. In this study, we reviewed the clinical and exome data of Filipino cochlear implantees and identified 12 novel variants in known genes for hearing loss and/or neurodevelopmental syndromes, as well as three candidate genes for hearing loss.

## 2. Materials and Methods

Out of our initial cohort of 30 Filipino patients, we previously identified a genetic variant as causal of hearing loss in 15 patients [32,33,34]. For this study, we reviewed the clinical records and temporal bone images of 15 Filipino cochlear implantees for whom no variants in known hearing loss genes were identified previously (Table 1) [34]. High-resolution computed tomography with 2–3 mm axial cuts and without contrast was performed using a Siemens Somatom Plus 4 CT Scanner in order to document temporal bone anomalies. DNA samples were submitted for exome sequencing at the University of Washington Northwest Genomics Center, as previously described [33,34]. The Roche NimbleGen SeqCap EZ Human Exome Library v.2.0 (~37 Mb target) was used for sequence capture, and sequencing was performed using an Illumina HiSeq to an average depth of 30 ×. Fastq files were aligned to the hg19 human reference sequence using Burrows-Wheeler Aligner, generating demultiplexed .bam files [35]. The Genome Analysis Tool Kit was used for realignment of indel regions (IndelRealigner), variant quality score recalibration (VQSR) and variant detection and calling, as well as generation of standard metrics used for quality control (QC) during exome analyses [36]. Low-quality and likely false-positive variants were flagged. The initial .vcf file for 29 *GJB2*-negative individuals included 82,853 variants, of which 74,965 passed QC filters. Variants from the entire .vcf file were annotated using ANNOVAR (annovar.openbioinformatics.org, last accessed March 18, 2021) [37]. Indels from the exome sequence data were also annotated using MutationTaster [38], however no rare or low-frequency variants were identified as potentially deleterious in the 15 patients studied.

Single nucleotide variants that passed QC were initially selected if they: (a) were homozygous or heterozygous in the 15 children with no known genetic etiology of hearing loss; (b) were stop, splice or missense variants; (c) had a minor allele frequency (MAF) <0.005 in any gnomAD (gnomad.broadinstitute.org, last accessed 31 March 2021), 1000 Genomes or Greater Middle East (GME) Variome population [4,39,40]; (d) from the Combined Annotation Dependent Depletion (CADD; cadd.gs.washington.edu, last accessed 31 March 2021) pre-computed scores database, had a scaled CADD score of ≥15 [41]; and (e) was predicted to be deleterious by at least one bioinformatics tool from dbSNFP41a [42]. Variants were excluded if they were common across our cohort, particularly if occurring within genes not previously associated with hearing loss but are found in multiple individuals that were identified to have variants in known genes for hearing loss [32,33,34]. This selection strategy resulted in a shorter list of 2570 variants, which was parsed further by prioritizing any variant that: (a) lies within a known hearing loss gene; (b) is a loss-of-function variant; (c) lies within a potentially novel gene but is homozygous or with two variants in the same gene in the same individual; and/or (d) lies within a gene that is identified in a mouse model with hearing loss. A list of 120 variants were rechecked against equivalent hg38 databases. Additional MAF checking was performed using the GenomeAsia 100K database (genomeasia100k.org, last accessed March 31, 2021) [43]. For known hearing loss genes, variants were ruled out if they occurred in a gene in which phenotypes are expressed only in homozygous or compound heterozygous individuals and the patient genotype is heterozygous. For the final list of 89 variants (Appendix A), the Integrative Genomics Viewer v2.8.3 was used to visualize variants from exome sequence data [44].

## 3. Results

Of the 15 children studied, six had EVA, three with high jugular bulb (HJB), two with SSCD/PSCD and two with malformed cochleae (Figure 1; Table 1). Five children had normal temporal bone CT/MRI images. From clinical history, seven children had exposures to infections and antibiotics, whether prenatally, at the neonatal stage or during early childhood (Table 1), suggesting that the previous infections or antibiotic use may have also played a role in their hearing loss etiology. Prior to CI, hearing loss in the 15 children was congenital, prelingual and severe-to-profound across frequencies, except for: (a) ID6 who had progressive hearing loss; and (b) ID20 who had fluctuating hearing loss with a steeply sloping audiogram and profound hearing loss at the high frequencies (Table 1).

A total of 21 rare/low-frequency potentially deleterious variants were identified in 17 genes (Table 1 and Table 2), all of which are known to be expressed in the mouse cochlea (gEAR, umgear.org, last accessed March 31, 2021). Although majority of the variants were heterozygous with likely autosomal dominant (AD) inheritance, several variants had seemingly different modes of inheritance, such as: (1) a homozygous *CLDN9* variant in ID20; (2) potentially compound heterozygous variants in *GDPD5, PCDH15* and/or *CDH23* in three children; and (3) an X-linked variant in *FLNA* in male patient ID24 (Appendix A). While our knowledge of mode of inheritance of these variants is limited, for five individuals the available history matches either an autosomal recessive (AR) pattern or AD inheritance with decreased penetrance (Appendix A). A more detailed genotype-phenotype correlation per gene and patient is hereby presented.

*DSPP:* Variants in *DSPP* (MIM 125485; 4q22.1) were first identified as a cause of AD hearing loss DFNA39 with dentinogenesis (MIM 605594) in Chinese families with dentinogenesis imperfecta 1 and adult-onset progressive sensorineural high-frequency hearing loss [45]. Additional hearing loss families, all of East Asian ethnicity, have been identified to have splice or missense variants within the first five exons of *DSPP* [46,47,48]. In the reported families, there was variability in age of onset, affected hearing frequencies, severity of hearing loss, and symptoms of tinnitus or balance problems [45,46,47,48]. In one family, the affected individuals had congenital hearing loss and bilateral cochlear defects with or without EVA [47]. In our study, patient ID1 had congenital hearing loss and unilateral EVA (Table 1; Figure 1). He was heterozygous for a novel variant c.730G>A (p.(Gly244Arg)), which lies within exon 4 of *DSPP* (Table 2). Although we have no record of dental abnormalities, he had small cysts identified in his brain MRI (Table 1). *Dspp* is expressed in inner ear, brain and pericytes of blood vessels in dental pulp of mice, and also in zebrafish otoliths [45,49,50]. He also has additional variants in *ANLN* (MIM 616027; 7p14.2)*, ZNF462* (MIM 617371; 9q31.2)*,* and *CEP290* (MIM 610142; 12q21.32). Each of these three genes harbor variants previously associated with hearing loss in various syndromes (Appendix A): branchio-otic syndrome with ossicular chain anomalies for *ANLN* [6]; Weiss-Kruszka syndrome with craniofacial dysmorphisms and developmental delay for *ZNF462* [51]; and Joubert syndrome with cerebral, retinal and kidney disease for *CEP290* [52]. While we cannot rule out if ID4′s brain cysts are related to these syndromic genes (e.g., kidney cysts are common in individuals with *CEP290* variants) [52], the other features of these syndromes are absent in patient ID1. Overall, the *DSPP* variant in ID6 fits his inner ear findings.

*LMX1A:* In addition to hearing loss, ID3 has malformed cochleae, left-sided EVA and global developmental delay (Figure 1; Table 1). Both the hearing loss and bony cochlear defects may be explained by novel heterozygous variants in two genes, namely *LMX1A* (MIM 600298; 1q23.3) c.606G>C (p.(Leu202Phe)) and/or *COL2A1* (MIM 120140; 12q13.11) c.3569G>A (p.(Arg1190His)) (Table 2 and Appendix A)*. LMX1A* is known for AD or AR nonsyndromic hearing loss [53,54], while *COL2A1* is related to Stickler syndrome type 1 with hearing loss (MIM 108300) as well as various skeletal phenotypes [55]. Homozygous *Lmx1a*-mutant mice lack endolymphatic ducts and have short cochlear ducts [56], which seem to recapitulate the incomplete cochlear turns and EVA in patient ID2. Additionally, hair cell loss and disorganization were seen in the cochleae of mutant mice [57]. However, unlike the deaf homozygous mice, the *Lmx1a*-heterozygous mice had normal hearing [56]. In contrast, two Dutch families with heterozygous missense *LMX1A* variants had mild-to-profound hearing loss of variable onset from infancy to adulthood [53]. On the other hand, a transgenic *Col2a1*-mutant mouse model had a smaller misshapen otic capsule as well as craniofacial abnormalities such as cleft palate and short mandible [58]; these latter features were not found in our patient ID3. In patient ID3, two variants in *USH2A* (MIM 608400; 1q41) were previously ruled out due to high MAF in the general Filipino population and lack of retinitis pigmentosa after years of follow-up (Appendix A). There were three other interesting variants in ID3 (Appendix A): (a) heterozygous missense variant in *ZFHX4* (MIM 606940; 8q21.13)–*ZFHX4* is one of two genes within the minimum region of overlap in patients with 8q21 microdeletions manifesting with intellectual and developmental disability, sensorineural hearing loss, craniofacial anomalies and hypotonia [59,60]; (b) heterozygous missense variant in *NRP1–*the *Nrp1^+/−^* mouse has abnormal auditory brainstem responses (ABR), progressive hearing loss, disorganized outer spiral bundles and enlarged microvessels of the stria vascularis [61]; and (c) a hemizygous missense variant in *ARHGAP4* (MIM 300023; Xq28), in which missense variants were previously described in children with intellectual disability [62,63]. This case shows potential overlap of clinical presentation due to multiple deleterious variants, of which the *LMX1A* variant is the strongest etiology of inner ear abnormalities in this patient while the *ZFHX4* or *ARHGAP4* variants may explain ID3’s developmental delay.

*DMXL2: DMXL2* (MIM 612186; 15q21.2) was recently identified to have missense variants causing AD nonsyndromic hearing loss in Chinese and Cameroonian families [64,65]. In these families, the affected individuals were mostly adult with progressive hearing loss and no reported temporal bone abnormalities, although one Cameroonian child had congenital profound hearing loss [65]. Our patient ID5 has a novel heterozygous *DMXL2* variant c.257T>C (p.(Leu86Ser)) (Table 2). In addition to prelingual profound hearing loss, her temporal bone CT showed a left HJB with evidence of dehiscence (Figure 1). She also had a history of neonatal infection as well as pervasive developmental delay (Table 1). In mice, cochlear expression of *Dmxl2* is limited to the hair cells and spiral ganglion neurons [64], and *Dmxl2*-knockout leads to preweaning lethality in the homozygous mouse and decreased bone mineral content if heterozygous (International Mouse Phenotyping Consortium (IMPC), www.mousephenotype.org, last accessed March 31, 2021). It is possible that the temporal bone findings are also an effect of the *Dmxl2* variant in ID5′s case. Biallelic loss-of-function *DMXL2* variants are also known to cause Ohtahara syndrome characterized by neurologic deficits including intellectual disability, developmental delay, hearing loss, polyneuropathy and also facial dysmorphisms [66]. However because patient ID5 only has a heterozygous *DMXL2* variant, the developmental delay may also be due to other causes, such as variants in *CCDC186* (MIM 619249; 10q25.3), *ZRF2* or *MCM3AP* (MIM 603294; 21q22.3) (Appendix A).

*PTPRQ: PTPRQ* (MIM 603317; 12q21.3) is a known cause of AD (MIM 617663) or AR (MIM 613391) nonsyndromic hearing loss in families and probands with multiple ethnicities, which may be variable in clinical presentation [67,68]. Patient ID6 is heterozygous for a novel missense variant c.6179T>C (p.(Val2060Ala)) within *PTPRQ* (Table 2). He also has progressive hearing loss, bilateral PSCD and HJB and right-sided EVA as temporal bone findings, as well as previous pneumonia and sinusitis (Table 1). In general previous reports of *PTPRQ*-related hearing loss excluded temporal bone anomalies, however narrowed internal auditory canals were found in a Chinese proband with compound heterozygous *PTPRQ* variants [69]. We previously ruled out a heterozygous variant in *TCOF1* (MIM 606847; 5q32-q33) due to lack of clinically diagnosed craniofacial hallmarks of AD Treacher-Collins syndrome (MIM 154500), but upon review, we cannot rule out that the *TCOF1* variant also contributes to hearing loss and temporal bone anomalies, as was previously described (Appendix A) [70]. Lastly a heterozygous variant in *DNAH14* (MIM 603341; 1q42.12), a candidate gene for primary ciliary dyskinesia and lung function in cystic fibrosis (Appendix A) [71,72], may play a role in ID6′s susceptibility to airway infections.

*PCDH15, CDH23* and *MYO7A:* While these three genes are known for Usher syndrome, they have AR nonsyndromic forms of hearing loss. In addition, *MYO7A* (MIM 276903; 11q13.5) variants may be inherited in an AD manner, while digenic inheritance for *PCDH15* (MIM 605514; 10q21.1) and *CDH23* (MIM 605516; 10q22.1) were demonstrated in mice and humans [73]. ID7 has hearing loss, HJB, mild motor delay, hypotonia, and urinary and upper respiratory infections (Figure 1; Appendix A). She has multiple variants of interest, but the strongest findings are compound heterozygous *PCDH15/CDH23* variants plus a heterozygous *MYO7A* variant (Table 2). Interestingly, the same *MYO7A* variant c.4921G>A (p.(Glu1461Lys)) is heterozygous in another patient ID18, who has nonsyndromic hearing loss (Table 2). This may suggest that the additional variants in ID7 contribute to her variable phenotype (Appendix A). Patient ID23 also has nonsyndromic hearing loss and three *CDH23* variants, however we could not confirm if these *CDH23* variants are compound heterozygous or inherited *in cis* due lack of available parental DNA (Table 2). These *CDH23* and *MYO7A* variants are reported as variants of unknown significance (VUS) in ClinVar (www.ncbi.nlm.nih.gov/clinvar/, last accessed 31 March 2021), while the *PCDH15* c.3787C>T (p.(Pro1263Ser)) variant is novel.

*COL11A1:* ID8 has hearing loss, left-sided SSCD, and heterozygous missense variants in two genes known for AD nonsyndromic hearing loss, namely *COL11A1* (MIM 120280; 1p21.1) and *TECTA* (MIM 602574; 11q23.3) (Table 1 and Table 2; Figure 1). Of the two deleterious variants, the *COL11A1* c.4364A>C (p.(Lys1455Thr)) variant is rarer (gnomAD EAS MAF = 0.0004). Previous reports on *COL11A1* or *TECTA* did not reveal inner ear abnormalities in patients with variants [74,75].

*IST1:* ID9 with profound hearing loss and left-sided EVA is heterozygous for a c.737C>G (p.(Pro246Arg)) variant in *IST1* (MIM 616434; 16q22.2). This rare deleterious variant (Table 2) was singled out due to a heterozygous *Ist1* mouse model that had abnormal ABR in early adulthood (IMPC). In mouse cochlea, *Ist1* is expressed in both hair cells and supporting cells (gEAR). Recently *de novo VPS4A* variants were identified to cause a multi-systemic neurodevelopmental disorder including sensorineural hearing loss due to the abnormal accumulation of IST1 protein in the limiting membrane of proband-derived fibroblasts and also in neuronal endosomes [76], suggesting that proper localization of IST1 is required for neuronal function. Taken together, our findings make *IST1* an excellent candidate gene for nonsyndromic hearing loss. Moreover, ID9 had poor CI outcomes, such as average CI-aided hearing threshold of 74 dB and speech tests using PEACH scores at 10–21%. Identification of additional patients with *IST1* variants is needed to verify these CI outcomes.

*SLC12A2:* ID13 who has hearing loss and global developmental delay is heterozygous for a novel stop variant c.2977G>T (p.(Glu993*)) in *SLC12A2* (also NKCC1; MIM 600840; 5q23.3) (Table 1 and Table 2). *SLC12A2* variants have been identified in patients with AD nonsyndromic hearing loss (MIM 619081), with AD Delpire-McNeill syndrome (MIM 619083), or AR Kilquist syndrome (MIM 619080). Recently McNeill et al. identified heterozygous *SLC12A2* variants in eight mostly pediatric patients with intellectual disability or developmental delay, and ~60% had bilateral sensorineural hearing loss [18]. Previous homozygous knockout of *Slc12a2* in mice led to loss of hearing and vestibular function, collapse of Reissner’s membrane, disorganization of the organ of Corti, and loss of hair cells and supporting cells [77]. On the other hand, heterozygous deletion of *Slc12a2* in mice resulted in early hearing loss that progressed with age despite normal inner ear morphology and histology [78].

*MYO18B:* Two patients had variants in *MYO18B* (MIM 607295, 22q12.1). Patient ID23 with nonsyndromic hearing loss has potentially compound heterozygous *CDH23* variants and also a novel heterozygous *MYO18B* variant c.1982G>A (p.(Trp661*)) (Table 2). The other patient ID19 has another novel variant c.2555C>T (p.(Ala852Val)) and severe cochleovestibular defects (Figure 1). In patient ID19, no other strong candidate variants or genes were identified (Appendix A). *MYO18B* variants were previously associated with autosomal recessive Klippel-Feil syndrome (MIM 616549) which is characterized by nemaline myopathy, facial dysmorphisms and hearing loss in up to 60% of patients [79]. Heterozygous *Myo18b*-knockout mice had abnormal ABR findings (IMPC), further supporting the role of heterozygous *MYO18B* variants in the etiology of hearing loss. Patients with hearing loss as part of Klippel-Feil syndrome were also diagnosed with inner ear dysplasias including internal acoustic canal deformities [80], which are similar to the temporal bone anomalies found in patient ID19 (Figure 1). Of the 30 Filipino patients, ID19 and ID23 who carry *MYO18B* variants had poorer outcomes after CI, with PEACH scores whether in quiet or noise at 4–37% despite average post-CI thresholds of ~40 dB at 0.25–8 kHz. This is not unexpected given potential cochlear nerve defects [30,31] that might not have been diagnosed radiologically (Figure 1). For ID19, her PEACH scores improved to >80% after 5 years of continued use of her implant on the left ear.

*FLNA:* The same *FLNA* (MIM 300017; Xq28) variant c.6350A>G (p.(Asn2117Ser)) that is classified as VUS in ClinVar was identified in two children ID20 and ID24 (Table 2). ID24 is male, hemizygous for the *FLNA* variant and has no other rare damaging variants in hearing loss genes. He is hemizygous for a known pathogenic variant in *G6PD* (MIM 305900; Xq28) which may explain his neonatal jaundice (Table 1 and Appendix A). *FLNA* is associated with multiple disorders, of which frontometaphyseal dysplasia (MIM 305620), Melnick-Needles syndrome (MIM 309350) and otopalatodigital syndrome (MIM 311300/304120) have been reported to include sensorineural hearing loss. ID24 has EVA in addition to the hearing loss but has no detailed assessment of additional skeletal anomalies; meanwhile temporal bone anomalies have been reported previously in a patient with Melnick–Needles syndrome [81]. On the other hand, the female patient ID20 who is heterozygous for the same *FLNA* variant has additional variants as the cause of hearing loss (Appendix A).

*CLDN9:* ID20 has fluctuating hearing loss at the high frequencies and additional sinonasal findings (Table 1). In addition to the *FLNA* variant, she is homozygous for a novel variant c.75C>G (p.(Cys25Trp)) in *CLDN9* (MIM 615799; 16p13.3) and heterozygous for *ANKRD11* (MIM 611192; 16q24.3) (Table 2 and Appendix A). KBG syndrome (MIM 148050) due to heterozygous *ANKRD11* variants manifests variably as macrodontia, intellectual disability and skeletal/craniofacial defects, including conductive or mixed hearing loss–these features do not fit the patient’s clinical presentation [82]. In contrast, a *CLDN9* frameshift variant was found in a Turkish family with AR nonsyndromic, progressive high-frequency hearing loss [83]; this clinical description is similar to that of ID20. In *Cldn9^-/-^* mice, defective tight junctions in the cochlea are hypothesized to cause the increased concentration K^+^ in the perilymph and massive hair cell loss [84]. In this case the sinonasal findings are probably not related to genetic susceptibility.

*GREB1L:* Previously variants in *GREB1L* (MIM 617782; 18q11.1-q11.2) were associated with AD nonsyndromic hearing loss with or without cochleovestibular malformations and non-ear phenotypes [20,21,22]. Our patient ID22 is heterozygous for a novel missense variant *GREB1L* c.3798C>G (p.(Ser1266Arg)) but has no other features in addition to profound hearing loss (Table 1, Table 2 and Appendix A). She also has a heterozygous variant in *CBLN3* (MIM 612978; 14q12). *Cbln3* is expressed in supporting cells and outer hair cells of the inner ear (gEAR), and also in the cerebellum and dorsal cochlear nucleus [85]. Heterozygous *Cbln3*-mutant mice have abnormal ABR (IMPC), implying that *CBLN3* is also a candidate gene for ID22′s hearing loss.

*GDPD5:* Patient 27 has two missense variants each in two genes: *GDPD5* (also *GDE2,* MIM 609632; 11q13.4-q13.5) which encodes an enzyme involved in glycerol metabolism; and *MADD* (MIM 603584; 11p11.2) (Appendix A). *Gdpd5* is expressed in hair cells and supporting cells of mouse cochlea (gEAR) and homozygous knockout mice have abnormal ABRs (IMPC). On the other hand, biallelic *MADD* variants cause a multisystemic neurodevelopmental disorder that includes sensorineural hearing loss in 17% of patients [86]. Our patient ID27 has hearing loss and bilateral EVA with no note of additional neurologic phenotypes (Table 1), suggesting that *GDPD5* is a candidate gene for her hearing loss.

## 4. Discussion

In this study, we identified novel variants in 14 genes: twelve are novel variants in eleven known hearing loss or neurodevelopmental genes *DSPP, LMX1A, DMXL2, PTPRQ, PCDH15, COL11A1, TECTA, SLC12A2, MYO18B, CLDN9* and *GREB1L*; while four variants are in candidate genes for hearing loss *IST1, CBLN3* and *GDPD5* (Table 2). In addition, several inner ear and temporal bone malformations were identified in variant carriers, namely: (1) EVA in carriers of *DSPP, IST1, FLNA* and *GDPD5* variants; (2) semicircular canal dehiscence in carriers of *DMXL2, PTPRQ* and *COL11A1/TECTA* variants; and (3) malformed cochleae in carriers of variants in *LMX1A* and *MYO18B* (Table 1; Figure 1). These findings suggest that at least some of these variants (e.g., variants in *DSPP, LMX1A* and *MYO18B*) are also potentially causal of temporal bone anomalies. Factors that may have contributed to an increased rate of variant identification from the sequence data of our cohort of 30 pediatric cochlear implant recipients include: (a) a more inclusive approach for low-frequency variants, particularly if the MAF was increased in an indigenous or isolated population which has high rates of intermarriage and potentially undiagnosed hearing loss (Table 2) [43,87,88]; and (b) genotype-phenotype correlation that takes into account additional clinical manifestations (e.g., developmental delay, recurrent infections) which overlap with features of syndromes or multi-systemic neurodevelopmental disorders. In the latter case, hearing loss might not be among the major criteria of the disorder, but the overall clinical presentation of the specific patient may fit previous descriptions of genotype-phenotype correlations that include hearing loss or bony defects.

Apparent contradictions in modes of inheritance may be due to undetected second variants for autosomal recessive disorders, which is a limitation of our study due to the lack of data on CNVs, cryptic splice sites, and non-coding regions [89]. Unfortunately, we only have DNA samples from patients and not from parents or additional relatives, so we cannot determine the identified variants’ pattern of inheritance or if they potentially arose de novo.

It is not unusual for the same gene to cause both autosomal dominant and autosomal recessive forms of hearing loss, e.g., *MYO7A* (MIM 276903) variants have been associated with either autosomal dominant (MIM 601317) or autosomal recessive non-syndromic hearing loss (MIM 600060), as well as autosomal recessive Usher syndrome type 1B (MIM 276900). Differences in modes of inheritance may be associated with phenotypic variability, such that variants known to cause autosomal recessive hearing loss that is characterized by prelingual profound hearing loss co-exist with heterozygous variants that cause autosomal dominant forms with milder hearing loss of later onset. Additionally, with the increasing number of identified genes for hearing loss, the occurrence of multiple variants within different genes that independently predispose to hearing loss in the same individual may be more common than previously thought [90]. Multiple variants in different genes may also contribute to variability in phenotypes (e.g., two genes with variants in the same individual causing different phenotypes rather than the same syndrome). An example would be ID24 in our cohort, in which a known pathogenic *G6PD* variant likely explains the patient’s neonatal jaundice, while the hearing loss is potentially due to a known variant in *FLNA.* Continued efforts in identifying novel genes mean that patient sequence data must be periodically reanalyzed not only to resolve a potential genetic etiology, but also to identify compound phenotypes due to variants in multiple genes. If multiple genes or variants are involved, additional studies on the functional effects per variant will aid in the determination of which variant is more strongly contributing to the hearing loss phenotype.

## 5. Conclusions

We identified novel variants in 11 known genes for hearing loss and neurodevelopmental phenotypes. We also present three genes *IST1*, *CBLN3* and *GDPD5* as potential candidate genes for hearing loss, all three of which have mouse models with abnormal ABR findings that are matched to the patient’s genotype. Our findings highlight the importance of identifying novel variants and genes in well-characterized patients from ethnic groups that are understudied for hearing loss.

## Figures and Tables

**Figure 1 genes-12-00566-f001:**
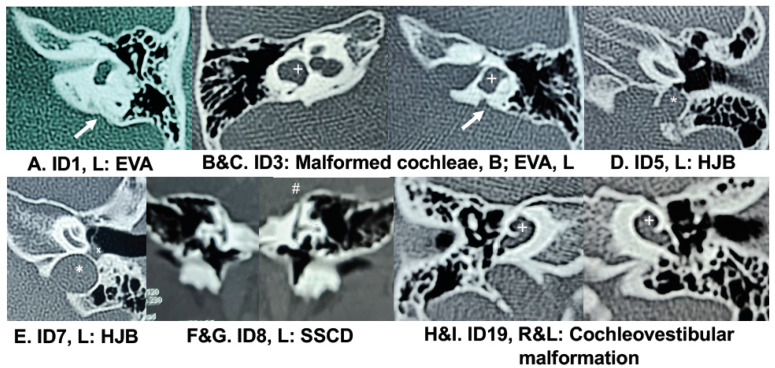
Temporal bone images in six patients with hearing loss. (**A**) ID1 with the heterozygous *DSPP* c.730G>A (p.(Gly244Arg)) variant has enlarged vestibular aqueduct (EVA, arrow) on the left. (**B**,**C**) ID3 with the heterozygous *LMX1A* and *COL2A1* variants has bilaterally malformed cochleae with incomplete cochlear turns (plus signs) and left-sided EVA (arrow). (**D**) ID5 with the heterozygous *DMXL2* variant has a high jugular bulb (HJB, asterisk) on the left. (**E**) ID7 with the heterozygous *MYO7A* variant plus potentially compound heterozygous *PCDH15* and *CDH23* variants has HJB (asterisk) on the left. There is also fluid in the middle ear space (marked by X), indicating otitis media. (**F**,**G**) ID8 with the heterozygous *COL11A1* and *TECTA* variants has left-sided superior semicircular canal dehiscence (SSCD, hash sign). (**H**,**I**) ID19 with the heterozygous *MYO18B* c.2555C>T (p.(Ala852)) variant has multiple congenital inner ear anomalies with bilaterally malformed cochleae, vestibules and semicircular canals (plus signs), as well as absence of the right cochlear and inferior vestibular nerves.

**Table 1 genes-12-00566-t001:** Clinical data for 15 Filipino children with hearing loss requiring cochlear implants (CI).

ID	Age at CI (yr)	Sex	Temporal Bone Findings	Clinical History	Gene
1	3.95	M	EVA, L	Bilateral small choroidal fissure cysts and a probable neuroepithelial cyst or prominent perivascular space involving the right peri-atrial white matter (MRI).	*DSPP*
3	2.83	M	Malformed cochleae with incomplete cochlear turns, B. EVA, L.	Global developmental delay	*LMX1A*
5	3.84	F	HJB with dehiscence, L	Prenatal antibiotic use for maternal respiratory infection. Patient used antibiotics in neonatal period for unspecified infection. Has pervasive developmental delay.	*DMXL2*
6	10.81	M	PSCD + HJB, B. EVA, R.	Pneumonia, sinusitis, and progressive hearing loss	*PTPRQ*
7	8.00	F	HJB, L. OM, L.	Mild motor delay and hypotonia. History of urinary and upper respiratory tract infections.	*MYO7A; PCDH15/CDH23*
8	3.03	M	SSCD, L	U/R	*COL11A1; TECTA*
9	8.19	F	EVA, L	Mother had urinary tract infection and eclampsia during pregnancy	*IST1*
13	5.95	M	Normal	Global developmental delay	*SLC12A2*
18	2.77	M	Normal	Sepsis and antibiotic/amikacin use during neonatal period	*MYO7A*
19	5.66	F	Malformed cochleae, vestibules and semi-circular canals, B. Absent cochlear and inferior vestibular nerves, R.	Maternal diabetes at 6 months gestation	*MYO18B*
20	14.59	F	Normal	Fluctuating hearing loss with steeply sloping audiogram prior to CI. Turbinate hypertrophy, allergic rhinitis, nasopharyngeal nodule.	*CLDN9*
22	4.40	F	Normal	U/R	*GREB1L; CBLN3*
23	4.61	F	Normal	U/R	*CDH23; MYO18B*
24	6.10	M	EVA, B	Fever, jaundice, foul umbilical discharge and apneic episodes with antibiotics and phototherapy in neonatal period	*FLNA*
27	7.72	F	EVA, B. OM, L.	U/R	*GDPD5*

M, male; F, female; U/R, unremarkable; B, bilateral; L, left; R, right; EVA, enlarged vestibular aqueduct; HJB, high jugular bulb; OM, otitis media; PSCD, posterior semicircular canal dehiscence; SSCD, superior semicircular canal dehiscence.

**Table 2 genes-12-00566-t002:** Novel variants and candidate genes ^1^ for hearing loss and temporal bone anomalies.

ID	Gene	Variant	rsID	gnomAD	GenomeAsia 100k SEA ^2^	Scaled CADD	Damaging Results from dbNSFP Tools
1	*DSPP*	*NM_014208: c.730G>A (p.(Gly244Arg))*	1044690454	NA	0.0014	24.3	FA,mLR,mSVM, MT,PP2,SI
3	*LMX1A*	*NM_177398: c.606G>C (p.(Leu202Phe))*	NA	NA	NA	24.8	FA,LRT,mLR, mSVM,MT,PP2, PR,SI
5	*DMXL2*	*NM_015263: c.257T>C (p.(Leu86Ser))*	761692429	OTH: 0.0005	NA	24.1	LRT,MT,PP2,SI
6	*PTPRQ*	*NM_001145026: c.6179T>C (p.(Val2060Ala))*	375150180	EAS: 0.00097	0.017	27.8	MT,SI
7	*PCDH15/CDH23*	*NM_001354411: c.3787C>T (p.(Pro1263Ser));* NM_022124: c.3262G>A (p.(Val1088Met))	775954124; 200632520	EAS: 0.004; EAS: 0.002	NA; 0.003	24.9; 24.3	MA,MT,PP2,PR, SI; LRT,MA,mLR, mSVM,MT,PP2,SI
23	*CDH23*	NM_022124: c.437C>T (p.(Pro146Leu)); c.3262G>A (p.(Val1088Met)); c.6911G>A (p.(Arg2304Gln))	765103490; 200632520; 201434373	NA; EAS: 0.002; EAS: 0.0015	0.001; 0.003; 0.007	24.7; 24.3; 22.7	LRT,MT,PP2,PR, SI; LRT,MA,mLR, mSVM,MT,PP2,SI; MT,SI
7, 18	*MYO7A*	NM_000260: c.4921G>A (p.(Glu1741Lys))	767975012	EAS: 0.0002	0.003	26.2	LRT,MT,PP2,PR
8	*COL11A1*	*NM_080629: c.4364A>C (p.(Lys1455Thr))*	769350133	EAS: 0.0004	NA	28.6	FA,LRT,mLR, mSVM,MT,PP2, PR,SI
8	*TECTA*	*NM_005422: c.2967C>A (p.(His989Gln)*	200821009	EAS: 0.003	0.0014	20.4	FA,LRT,mLR, mSVM,MT,PP2, PR,SI
9	***IST1***	**NM_001270976: c.737C>G (p.(Pro246Arg))**	774343604	EAS: 0.0002	NA	24.0	LRT,MT,PP2,PR, SI
13	*SLC12A2*	*NM_001046: c.2977G>T (p.(Glu993*))*	NA	NA	NA	60.0	MT
19	*MYO18B*	*NM_032608: c.2555C>T (p.(Ala852Val))*	NA	NA	NA	26.1	FA,LRT,mLR, mSVM,MA,MT, PP2,PR,SI
23	*MYO18B*	*NM_032608: c.1982G>A (p.(Trp661*)*	372939044	AFR: 0.0005	NA	44.0	LRT/MT
20	*CLDN9*	*NM_020982: c.75C>G (p.(Cys25Trp))*	368045321	OTH: 0.0005	0.004	20.6	FA,LRT,MA,mLR,mSVM,MT,PP2, PR,SI
20, 24	*FLNA*	NM_001110556: c.6350A>G (p.(Asn2117Ser))	375205247	EAS: 0.002	NA	20.2	FA,LRT,MT,PR
22	*GREB1L*	*NM_001142966: c.3798C>G (p.(Ser1266Arg))*	954005555	EAS: 0.0006	0.003	16.6	LRT,MA,MT,PR, SI
22	***CBLN3***	**NM_001039771:** **c.550C>T** **(p.(Arg184Cys))**	562291434	EAS: 0.0002	NA	32.0	LRT,MT,PP2,PR, SI
27	***GDPD5***	**NM_030792: c.554G>A (p.(Arg185His)); c.404C>T (p.(Thr135Met))**	745585758; 373413383	ME: 0.003; AFR: 0.00002	0 (South Asia = 0.0007); NA	23.1; 24.8	LRT,MT,PP2; LRT,MA,MT,PP2

^1^. Bold font denotes candidate genes, while novel variants in known genes are in italics. ^2^. Variants identified in the Southeast Asian (SEA) population in the GenomeAsia 100k database were mostly from individuals of Filipino (*n* = 52) or Indonesian (*n* = 68) descent. MAF from Filipino alleles were identified in indigenous Negrito (Ati, Aeta) tribes, which are usually intermarried and are not representative of the general Filipino population. NA, not available/found; EAS, East Asian; AFR, African; ME, Middle Eastern; OTH, other; FA, FATHMM; LRT, likelihood ratio test; mLR, meta-logistic regression; mSVM, meta-support vector machine; MA, MutationAssessor; MT, MutationTaster; PP2, PolyPhen2; PR, PROVEAN; SI, SIFT.

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
