# Peer review of "Identification of Novel Candidate Genes and Variants for Hearing Loss and Temporal Bone Anomalies"

_genes, 2021, doi:10.3390/genes12040566_

Round 1

Reviewer 1 Report

Dear Authors,

compliment for your research work and new suggestions. I can not figure it out, how you did the genetical testing in  only 15 patients? I would understand if they would all have temporal bone malformations - but 5 of then had not. Hot come that your Ci program consists of only 30 patients? Time frame is needed. If I understanded your sentences regarding your patients numbers correctly. Please, clarify this. And than you concudluded that your "success rate" for finding a cause of deaf is 77%? I do not think you can conclude this "percentage result" on 30 patients.

Author Response

Reviewer 1:

Dear Authors, compliment for your research work and new suggestions. I can not figure it out, how you did the genetical testing in  only 15 patients? I would understand if they would all have temporal bone malformations - but 5 of then had not. Hot come that your Ci program consists of only 30 patients? Time frame is needed. If I understanded your sentences regarding your patients numbers correctly. Please, clarify this. And than you concudluded that your "success rate" for finding a cause of deaf is 77%? I do not think you can conclude this "percentage result" on 30 patients.

Response:  Thank you for the compliment. The cohort of 30 CI patients was recruited before our first publication on GJB2 (Chiong et al., Audiol Neurotol Extra, 2013). Because of limited funding from the Philippines, there has not been additional recruitment since. To clarify this situation, this statement was added as first sentence in paragraph 1 of Materials & Methods, transitioning into the second sentence with an added clause: “Out of our initial cohort of 30 Filipino patients, we have identified a genetic variant as causal of hearing loss in 15 patients.20-22 For this study, we reviewed the clinical records and temporal bone images of fifteen Filipino cochlear implantees for whom no variants in known hearing loss genes were identified previously (Table 1).22” Additionally the mention of 77% success rate was removed from Discussion. Sentence 4, paragraph 1 of Discussion was revised as follows: “Our current work increased the number of pathogenic or potentially pathogenic variants identified from the sequence data of our cohort of 30 pediatric cochlear implant recipients.20-22

Reviewer 2 Report

In this study, authors review the possible genetic implications on a cohort of temporal bone-damaged Filipino cases with no hearing loss-related variant described. The results are strong and can be useful to pinpoint new candidate genes related with temporal bone abnormalities and hearing loss not considered before.

The main point of interest of this study is the use of a specific ethnic group to search possible novel genes understudied in other hearing loss genetic studies. Also, the determination of possible compound inheritance defining hearing loss phenotypes can be useful to understand the differences between different hearing loss clinical populations.

A technical limitation of the study is the sequencing design due to exome sequencing only covering a little part of the entire genome for novel variation discovery instead of whole-genome sequencing. However, I understand this is an expensive approach to aboard without previous knowledge of the genetic background of this cohort.

Some minor points of mention:

  1. Why authors use Hg19 reference genome for their alignments? Currently, Hg38 is vastly used for most of genome studies. Also, reference frequency databases as gnomAD are moving directly to Hg38 with several more samples, changing perhaps the MAF values a bit for certain variants.
  2. Authors should disclose which quality filtering methods they used in their variant calling pipeline. VSQR is the standard methodology suggested by GATK but other quality filtering parameters and methods could be used. For replicability matters, I suggest authors to include this information in methodology paragraph.
  3. I consider a sentence detailing why this low MAF threshold was used instead of a more common MAF threshold is needed in the methodology instead of discussion. In fact, the filter used is quite limited for the condition studied and, though the observed cohort could be difficult to study due to its inbreeding ethnical characteristics, it should be addressed. Also, some prevalence data regarding hearing loss + temporal bone abnormalities such EVA or PSCD could be useful to understand the impact in this selected Asian population.
  4. Did any of the recruited cases report familial history? It seems this is not addressed anywhere in the text and it could lead to certain inheritance conclusions of interest. This could be also added to Table 1 if obtained.
  5. I would like to suggest adding a clear workflow for the candidate variant filtering from the total number of variants called to the chosen candidate ones at the end of the filtering. CADD was mainly used for pathogenicity prediction, while other annotations were barely used and only noted in the tables. Perhaps they can be used to prioritize those potentially more damaging according to the number of tools prediction pathogenicity for a given variant.
  6. Authors point the difficulty to check CNVs in their exome data. However, CNVs can be checked easily with tools currently used for germline analysis such CNVkit or CNVnator. Though this will only obtain CNVs overlapping exons, this could add another layer of complexity and relevant information to this study.  I understand this could lead to a major change for the authors but I would like to understand why this wasn't even tried.

Author Response

Reviewer 2:

In this study, authors review the possible genetic implications on a cohort of temporal bone-damaged Filipino cases with no hearing loss-related variant described. The results are strong and can be useful to pinpoint new candidate genes related with temporal bone abnormalities and hearing loss not considered before.

The main point of interest of this study is the use of a specific ethnic group to search possible novel genes understudied in other hearing loss genetic studies. Also, the determination of possible compound inheritance defining hearing loss phenotypes can be useful to understand the differences between different hearing loss clinical populations.

A technical limitation of the study is the sequencing design due to exome sequencing only covering a little part of the entire genome for novel variation discovery instead of whole-genome sequencing. However, I understand this is an expensive approach to aboard without previous knowledge of the genetic background of this cohort.

Response: Thank you for the positive comments. We only had a limited amount of funding, hence only exome sequencing was possible.

Some minor points of mention:

Why authors use Hg19 reference genome for their alignments? Currently, Hg38 is vastly used for most of genome studies. Also, reference frequency databases as gnomAD are moving directly to Hg38 with several more samples, changing perhaps the MAF values a bit for certain variants.

Response: Thank you for this helpful comment. All entries in Table 2 and Table S1 were checked against hg38 databases and updated accordingly. Database versions were added to Materials & Methods, paragraph 1.

Authors should disclose which quality filtering methods they used in their variant calling pipeline. VSQR is the standard methodology suggested by GATK but other quality filtering parameters and methods could be used. For replicability matters, I suggest authors to include this information in methodology paragraph.

Response: The following sentences were added to paragraph 1 of Materials & Methods: “The Roche NimbleGen SeqCap EZ Human Exome Library v.2.0 (~37Mb target) was used for sequence capture, and sequencing was performed using an Illumina HiSeq to an average depth of 30×. Fastq files were aligned to the hg19 human reference sequence using Burrows-Wheeler Aligner, generating demultiplexed .bam files.28 The Genome Analysis Tool Kit was used for realignment of indel regions (IndelRealigner), variant quality score recalibration (VQSR) and variant detection and calling, as well as generation of standard metrics used for quality control during exome analyses.29 In the generated .vcf file, low-quality and likely false-positive variants were flagged. The initial .vcf file for 29 GJB2-negative individuals included 82,853 variants, of which 74,965 passed QC filters. Variants from the entire .vcf file were annotated using ANNOVAR.30 Indels from the exome sequence data was also annotated using MutationTaster,31 however no rare or low-frequency variants were identified as potentially deleterious.”

I consider a sentence detailing why this low MAF threshold was used instead of a more common MAF threshold is needed in the methodology instead of discussion. In fact, the filter used is quite limited for the condition studied and, though the observed cohort could be difficult to study due to its inbreeding ethnical characteristics, it should be addressed.

Response: The data was reanalyzed using updated database information and a MAF threshold of 0.005 in any gnomAD, 1000 Genomes or GME Variome population. This MAF threshold of 0.005 follows the recommendations of Azaiez et al., 2018, particularly for autosomal recessive hearing loss. The reanalyses resulted in a larger number of variants (n=2,570). We then prioritized any variant that (a) lies within a known hearing loss gene, (b) is a loss-of-function variant, (c) lies within a potentially novel gene but is homozygous or with two variants in the same gene in the same individual, and/or (d) lies within a gene that is identified in a mouse model with hearing loss. This exercise resulted in the identification of new variants that were not previously included in our annotation files or had a different MAF, and these variants are now added to the manuscript.

Also, some prevalence data regarding hearing loss + temporal bone abnormalities such EVA or PSCD could be useful to understand the impact in this selected Asian population.

Response: A short paragraph on this matter was added to Discussion.

Did any of the recruited cases report familial history? It seems this is not addressed anywhere in the text and it could lead to certain inheritance conclusions of interest. This could be also added to Table 1 if obtained.

Response: We now include details on family history in Table 1.

I would like to suggest adding a clear workflow for the candidate variant filtering from the total number of variants called to the chosen candidate ones at the end of the filtering. CADD was mainly used for pathogenicity prediction, while other annotations were barely used and only noted in the tables. Perhaps they can be used to prioritize those potentially more damaging according to the number of tools prediction pathogenicity for a given variant.

Response: We now specify in Materials & Methods (paragraph 1, sentence 11) that a deleterious prediction in at least one bioinformatics tool is used to select variants. Please also see answer to comment 3 above.

Authors point the difficulty to check CNVs in their exome data. However, CNVs can be checked easily with tools currently used for germline analysis such CNVkit or CNVnator. Though this will only obtain CNVs overlapping exons, this could add another layer of complexity and relevant information to this study.  I understand this could lead to a major change for the authors but I would like to understand why this wasn't even tried.

Response: We previously had some experience with CNV calling using exome data from families with hearing loss with valuable assistance and collaborative work from experts in the field, and it was not helpful at all due to the multiple spotty regions which ended up being false-positives after validation testing. This effort took up a lot of time with no yield; on the other hand, re-analyses identified other SNVs or indels that were previously missed in the same exomes that were analyzed for CNVs. Our group would rather get additional funding to recruit new families and also perform whole-genome sequencing which will have much better yield with CNVs.

Reviewer 3 Report

The manuscript by Santos-Cortez et al: Identification of a novel candidate genes and variants for hearing loss and temporal bone anomalies, review clinical and exome data from 15 children with hearing loss. They found 9 that were heterozygous for rare, damaging novel missense variants in eight genes. Two children with hearing loss carried some variants. Authors concluded the is very important to identify novel variants and genes in ethnic groups that are understudied for hearing loss. In this study DNA samples from the patients were submitted for exome sequencing. Novel variants in eight genes: five are novel variants in known hearing loss, neurometabolic or immune genes and three candidate genes for hearing loss. Abstract: well written to the point Introduction: pertinent and contemporary Methods: well described. Please mention how the images for figure 1 were collected prepared. Results: Authors described systematically their findings Discussion and conclusions: Contrast and compare their results with previous literature Figures and table : Organized. Bibliography: Contemporary and inclusive.

Author Response

Reviewer 3:

The manuscript by Santos-Cortez et al: Identification of a novel candidate genes and variants for hearing loss and temporal bone anomalies, review clinical and exome data from 15 children with hearing loss. They found 9 that were heterozygous for rare, damaging novel missense variants in eight genes. Two children with hearing loss carried some variants. Authors concluded the is very important to identify novel variants and genes in ethnic groups that are understudied for hearing loss. In this study DNA samples from the patients were submitted for exome sequencing. Novel variants in eight genes: five are novel variants in known hearing loss, neurometabolic or immune genes and three candidate genes for hearing loss. Abstract: well written to the point Introduction: pertinent and contemporary Methods: well described. Please mention how the images for figure 1 were collected prepared. Results: Authors described systematically their findings Discussion and conclusions: Contrast and compare their results with previous literature Figures and table : Organized. Bibliography: Contemporary and inclusive. 

Response: The positive comments are very much appreciated. This sentence was added to Materials & Methods: “High-resolution computed tomography with 2-3 mm axial cuts without contrast was performed using a Siemens Somatom Plus 4 CT Scanner in order to document temporal bone anomalies.”